# Calcium and Potassium Accumulation during the Growing Season in Cabernet Sauvignon and Merlot Grape Varieties

**DOI:** 10.3390/plants11121536

**Published:** 2022-06-08

**Authors:** Eleonora Nistor, Alina Georgeta Dobrei, Giovan Battista Mattii, Alin Dobrei

**Affiliations:** 1Department of Horticulture, Banat University of Agricultural Sciences and Veterinary Medicine, 300645 Timisoara, Romania; nisnoranisnora@gmail.com (E.N.); ghitaalina@yahoo.com (A.G.D.); 2Department of Agriculture, Food, Environment and Forestry (DAGRI), University of Florence, 50144 Florence, Italy; giovanbattista.mattii@unifi.it

**Keywords:** Ca, K, grapevine, environment, accumulation, flesh, skin, seeds

## Abstract

The evolution of calcium (Ca) and potassium (K) accumulation in grape berries during the growing season provided information on the productivity and quality of grape crops, considering that both elements have numerous physiological effects. The aim of the study was to determine and compare Ca and K accumulation in berries from ‘Cabernet Sauvignon’ and ‘Merlot’ grape varieties influenced by the number of days after flowering (DAF) over three consecutive growing seasons (2019–2021) in Recaş vineyards, from Banat Region in Western Romania. Results showed that Ca accumulation in the berries continued at slow rates after veraison when water was available for both varieties; accumulation was observed mainly in the skin, suggesting translocation from the flesh. Regression analysis showed a strong dependence on the interval of 65–75 DAF for Ca accumulation in the skin. K accumulation increased after the onset of veraison until 70 to 90 DAF in both varieties, with higher accumulation in the flesh than in the skin. No significant differences were found among varieties regarding the Ca and K content during the study period. In both varieties, the relationship between sugar accumulation and the Ca/K content was highly significant. Given the importance of both elements, new data may contribute to establishing the optimum grape ripeness in relation to the sugar concentration in the berries.

## 1. Introduction

Almost all minerals have either positive or negative influences upon enzymes within organisms [1]. Either too low or too high concentration in a plant’s nutrition may potentially cause deficiencies or diseases [2]. However, macronutrients in soil solution can reduce the toxicity of some chemical components or they can favorably influence nutrient assimilation by the plants [3].

The role of Ca in plants is not fully defined, but its involvement in the firmness of cell walls [4] and grape berry set (defined as cell division and expansion) or berry ripening (defined as softening) is well researched [5,6,7]. Ca is one of the most versatile components in plants and it is involved in responses to biotic and abiotic stimuli such as low and high temperature, light, drought, and osmotic stress in nearly all stages of plant development [8]. By investigating the influence of Ca on vines and berry development, physiological symptoms for the level of plant nutrition may be identified, as well as the possibility of improving berry quality [9]. Ca is absorbed through the roots, and it reaches the plant and berries through the xylem and apoplast [9,10] due to low phloem mobility [10]. Some studies have shown that Ca accumulation ceases prior to veraison [7,11,12,13], while others state that this accumulation continues during ripening, especially in the seeds, increasing the total Ca concentration in the berries [7,14,15,16,17]. The Ca content influences not only flesh firmness but also water supply, ripening and heat stress response [18]. It delays senescence if it is at an optimal concentration and it increases resistance to Botrytis cinerea [19,20,21,22]. Ca deficiency induces physiological disorders in berries, such as dehydration (shriveling), and it delays ripening [23].

Among the nutrients, K is important for stimulation of berry formation and development, sugar accumulation, and wood maturation. K uptake through the roots is higher than most other elements [24].

K plays multiple roles in plants. It is involved in the activation of more than 60 different enzymes that serve as catalysts for biochemical reactions. It also has the role of neutralizing the pH of plant cells, to ensure an optimal level (between 7 and 8) for chemical reactions [2]. K is important in photosynthesis: plants use the sun’s energy to combine water and CO_2_ in order to produce sugars, resulting in adenosine triphosphate (ATP)—the energy-rich compound in which K plays a complex role (K deficiency reduces ATP synthesis and photosynthesis) [25]. K is also essential for photosynthesis (because it regulates the closing and opening of stomata) and for the transport of nutrients and water, including plant cooling [26,27]. K contributes to drought resistance by improving water use efficiency, resulting in improved stomatal function. Furthermore, K promotes grape setting, and it also accelerates ripening [28]. In conditions of low K levels, pores react slowly and sometimes open/close every few hours instead of minutes, resulting in water loss, which renders the vine prone to water stress [28]. Given that ATP is involved in sugar transport as an energy source, insufficient K prevents the normal functioning of this process [29]. The regulation of plant growth processes based on cellular proteins and enzymes depends on the presence of K, and low K levels hinder protein synthesis [30]. Since starch synthesis also depends on K levels, if the K content is too low, nitrogen compounds accumulate in excess, to the detriment of starch formation [31]. However, a high K content favors the absorption of iron, which influences aroma, taste, color, and wine aging, or it may create conditions leading to grape collapse (physiological K/Mg imbalance in berry cells) [32]. Therefore, K deficiency, especially in key phases of vine development, may reduce resistance to disease, which may cause significant damage [33]. In addition, high K content reduces free acids, whereas the overall pH increases [34].

The purpose of this study was to quantify and compare the accumulation of K and Ca in berries of grape varieties ‘Cabernet Sauvignon’ and ‘Merlot’. The study aimed to determine the highest Ca and K accumulation in grape berries depending on DAF, so as to enable viticulturists to apply fertilizing solutions to produce the best possible wine. Considering the importance of Ca and K for berry ripening, the newly collected data provide evidence regarding the evolution of these components in order to determine optimal grape ripeness (as yet theoretically, given that it is largely dependent on the desired type of wine, while being directly related to variety, weather conditions, climate, and terroir), in correlation with the sugar concentration of the berries.

## 2. Results

The 2019 spring was cold, especially overnight, delaying the vines’ vegetative and reproductive development by about two weeks. Some varieties were more affected than others; for example, ‘Merlot’ started flowering a week later (5 May 2019) than ‘Cabernet Sauvignon’ completed the flowering process. However, there were many disturbances in May due to heavy rainfall (259.3 mm/m^2^). There was also a lot of moisture in June (154.7 mm/m^2^) and July (113.8 mm/m^2^). Warmer weather in early June prompted the vines to start recovering from the delay in vegetation.

The mild winter of 2020, followed by a cold and very dry April, with temperature fluctuations, increased the risk of frost damage in the buds. Excessive drought was followed by heavy rainfall in June and July; drought also occurred in autumn during harvesting.

The 2021 growing season was atypical, with extremely cold and rainy weather throughout spring until the end of May; it was hot and dry from June to October, except for July, when there was heavy rainfall. Under these circumstances, the vine sprouted 2–3 weeks later, and this lag was maintained until autumn, thus delaying harvesting. Fortunately, the late budding of vines in the Recaș vineyard kept the grape harvest from being damaged by the hoarfrost in early April, which damaged many vineyards in Europe.

Flowering occurs between 25 May and 5 June in ‘Cabernet Sauvignon’ and 22 May to 3 June in ‘Merlot’ depending on the weather in the research area. Veraison usually takes place between 60 and 65 DAF, on 25–30 July in the case of ‘Cabernet Sauvignon’ and 22–27 July in ‘Merlot’. Although ‘Cabernet Sauvignon’ usually blooms earlier, it develops more slowly during the growing season.

### Ca and K Accumulation during Berry Development

To better illustrate the Ca and K accumulation in grape berries, a polynomial regression (second order equations) was calculated. In the berry flesh, Ca accumulated progressively until veraison and then slowly decreased until maturity, as shown by the line following the data, with a medium or small R^2^ value (2019–2021; Figure 1).

Ca accumulation in the skin and flesh of ‘Merlot’ and ‘Cabernet Sauvignon’ berries was very significantly affected by DAF (*p* < 0.0001; α = 0.05; F (4, 40) = 64.29). There were no significant differences among varieties in Ca accumulation at different DAF (*p* > 0.9999). Similar results were found for the other two growing seasons (2020 and 2021).

Based on the regression, the Ca concentration was highly correlated with DAF in both varieties, indicating high data coherence. In the ‘Cabernet Sauvignon’ variety, maximum Ca accumulation in berry flesh occurred at 62, 57, and 51 DAF in 2019, 2020, and 2021, respectively. In the ’Merlot’ variety, maximum Ca accumulation occurred at 50, 65, and 57 DAF. The maximum Ca accumulation in the skin occurred later than in the flesh. The maximum Ca accumulation in the skin of ’Cabernet Sauvignon’ berries was 90 DAF in 2019, 89 DAF in 2020, and 75 DAF in 2021. For the ’Merlot’ variety, maximum Ca accumulation in berry skin occurred at 85 DAF in 2019 and 2020 and 77 DAF in 2021. The two varieties responded differently to temperature and moisture conditions, with ’Cabernet Sauvignon’ being more resistant to stressors, as compared to the other variety.

K accumulation displays the same pattern as Ca in all three growing seasons (2019–2021) during berry development. The amount of K increased steadily after a few days from veraison onset, until 70 to 90 DAF in both ‘Cabernet Sauvignon’ and ‘Merlot’, when the accumulation was higher in the flesh than in the skin (Figure 2). During the three growing seasons, K accumulation in the berry skin and flesh of both varieties was strongly influenced by DAF (*p* < 0.0001; α = 0.05; F (4, 60) = 92.71). No significant differences were found between varieties regarding the accumulation of K in the skin and flesh of berries from both varieties at different DAF intervals (*p* > 0.9999).

In the flesh of ’Merlot’ berries, K content reached its maximum 94 DAF in 2019 and 2020 and 88 DAF in 2021. The maximum K accumulation in the skin of ‘Cabernet Sauvignon’ berries ranged from 84 DAF in 2019 to 89 DAF in 2020 and 85 DAF in 2021. In the ’Merlot’ variety, the maximum K content in the skin was also variable, depending on the year correlated with DAF (at 81 DAF in 2019, at 89 in 2020 and 87 in 2021).

According to the Tukey test, both Ca and K accumulation were influenced significantly by DAF in ‘Merlot’ and ‘Cabernet Sauvignon’ varieties (with *p* < 0.0001; α= 0.05; F= 89.26). In both varieties, the presence of K had no effect (*p* > 0.9999) on Ca accumulation (or vice versa). There were no significant differences among varieties (*p* > 0.9997) regarding the amounts of Ca and K accumulated in berries, at different DAF intervals, for the 2019 growing season. Results for the 2020 and 2021 growing seasons were similar (Figure 3A–C).

Ca accumulation in the seeds seemed to display a similar pattern before and after the veraison period, but the amount was dissimilar for the same period in the different growing seasons (Figure 4A–C). It was found that in the seeds of variety ‘Cabernet Sauvignon’, the maximum Ca accumulation occurred earlier than in ‘Merlot’ each year. In ‘Cabernet Sauvignon’, the maximum Ca accumulation occurred 60 DAF in 2019, 55 DAF in 2020, and 65 DAF in 2021. In ‘Merlot’, maximum seed enrichment occurred later, 65 DAF in 2019, 67 DAF in 2020, and 66 DAF in 2021. In terms of K content, the maximum in 2019 was 69 DAF in ‘Cabernet Sauvignon’ and earlier (66 DAF) in ‘Merlot’. In 2020 and 2021, the maximum seed K content was reached later in ‘Merlot’ (80 and 70 DAF, compared to 75 and 69 for ‘Cabernet Sauvignon’), when climatic conditions were more favorable in August (the veraison phase) compared to 2019.

The amounts of Ca and K accumulated in the seeds of the two varieties were very significantly influenced by DAF during the three growing seasons. However, no differences were found between varieties regarding Ca or K accumulation in seeds at different DAF intervals.

During the 2019 growing season, the interdependence between sugar and Ca accumulation in ‘Cabernet Sauvignon’ berries was highly significant (*p* < 0.0001; α = 0.05). The correlation between sugars and K accumulation in berries and the effect of DAF on sugar accumulation in berries were both statistically significant (*p* = 0.0054; α = 0.05) (Figure 5A).

Sugar accumulation was very significantly influenced (*p* = 0.0074) by DAF in the variety ‘Merlot’ during the same growing season (Figure 5B). The relationship between sugar and Ca accumulation in Merlot berries was highly significant (*p* < 0.0001; α = 0.05), while the relationship between sugar and K accumulation was statistically significant (*p* = 0.067).

The effect of DAF on sugar accumulation in ‘Cabernet Sauvignon’ berries was statistically significant in the 2020 growing season (*p* = 0.0099; =0.05). There was also a highly significant relationship between sugar content and both Ca/K accumulation (*p* < 0.0001; α = 0.05).

The effect of DAF on sugar accumulation in ‘Merlot’ berries in the growing season 2020 was lower but statistically significant (*p* = 0.0112; α = 0.05). However, the relationship between sugar accumulation and Ca/K content at different DAF was highly significant (*p* < 0.0001; α = 0.05) (Figure 5C,D).

A less significant influence of DAF upon sugar accumulation in ‘Cabernet Sauvignon’ berries (*p* = 0.0106; α = 0.05) was observed during the 2021 growing season. The relationship between sugar and Ca accumulation in berries was highly statistically significant (*p* < 0.0001; α = 0.05); the influence of sugar on berry K content was smaller but statistically significant (*p* = 0.059; α = 0.05). Similar results were found in the variety ‘Merlot’ in 2021—DEF had less effect on sugar content in the berries (*p* = 0.0122; = 0.05). The Ca content in berries was strongly statistically related to the sugar content (*p* < 0.0001; α = 0.05), but K and sugars were less significantly correlated (*p* = 0.063; = 0.05) (Figure 5E,F).

## 3. Discussion

According to the research results, the variety ‘Cabernet Sauvignon’ accumulated more Ca in the berry flesh from veraison to maturity than the variety ‘Merlot’. Although Ca accumulation in the flesh was higher in both varieties, the regression coefficient was moderate in ‘Cabernet Sauvignon’ and high in ‘Merlot’.

However, the regression curves show that the pre-veraison Ca content in the flesh was higher in the variety ‘Merlot’ in the 2020 and 2021 growing seasons. In addition, Ca accumulation in the berry flesh after veraison was higher in the variety ‘Merlot’ compared to the berry flesh of ‘Cabernet Sauvignon’ in the 2021 growing season. Ca accumulation after veraison in 2020 was higher in the berry skin of ‘Cabernet Sauvignon’ than the ‘Merlot’. In the last growing season, the Ca content in the skin was balanced for both varieties. Irrespective of the variety, Ca accumulation during berry development indicated the migration of Ca from the flesh to the skin.

In 2019, K accumulation in the flesh and skin of the berries was similar in both varieties (favoring ‘Cabernet Sauvignon’). It displayed a constant, slightly increasing trend in the first phenological stages, influenced by the cold and rainy weather in May and high temperature in June. In contrast to 2019, in the 2020 growing season, the unbalanced humidity, in combination with the average temperature, influenced the accumulation of K in the whole berries of both varieties. Overall, ‘Cabernet Sauvignon’ accumulated much more K in the flesh than ‘Merlot’, but it should be noted that after 70 DAF, the ‘Merlot’ variety recorded higher enrichment in the skin as compared to ‘Cabernet Sauvignon’, indicating a better adaptation to temperature and rainfall variability during the veraison period (June was one of the rainiest months in the last 60 years, followed by a hot and rainy month of July and a warm and dry month August), especially at the microclimate level. In 2021, the K content in the flesh and whole berries was lower, as a result of the high temperatures throughout the growing season.

Mohammad [35] found the highest rates of Ca accumulation in *Vitis vinifera* L. cv. Asgari at 60 DAF, with accumulation ceasing after 80 DAF. The accumulation of Ca and K in the grape berries was influenced by climate variability and certainly by water availability in each growing season. Flowering and veraison were delayed by low temperatures, especially at night and heavy rainfall in spring 2019. K accumulation in the berries increased steadily before and after veraison, while Ca accumulation increased more rapidly, especially in the first phase of berry development. There was a slight difference in K accumulation between the varieties, with ‘Cabernet Sauvignon’ displaying slightly higher levels.

Regarding K accumulation, the regressions described the relationship effectively (R^2^ above 0.80). In the 2019 growing season, Ca accumulation reached the highest value between 70 and 90 DAF, while K levels increased after 80 DAF. In 2020, the increase in Ca accumulation was observed 60 days DAF. Similar to Ca, K accumulation was also higher and more intense 80 DAF. A similar pattern was observed for K in the 2021 growing season, but Ca accumulation in higher amounts started earlier (approximately 40 DAF). Lower accumulation was observed for both components and varieties in the hot 2021 growing season. ‘Cabernet Sauvignon’ displayed higher Ca and K accumulation than ‘Merlot’, except for the 2021 growing season, when ‘Merlot’ displayed higher accumulation at maturity.

The seeds are a less significant sink of Ca and K than the flesh or skin, but they add up to the total berry content. In all three growing seasons, K accumulation in the seeds before veraison was slightly more evident in ‘Cabernet Sauvignon’. After veraison, the variety ‘Merlot’ accumulated more K, indicating a better adaptation to temperature stress. Each year, high temperature variability between day and nighttime (diurnal temperature range) was recorded, especially in spring. It is well known that temperature significantly impacts bud break, the flowering stage and veraison, as well as grape berry metabolism and quality composition. Cohen et al. [36] also reported that diurnal temperatures have a major influence on grape berry development and components. The accumulation and content of Ca and K in the structure of grape berries is influenced by soil nutrients and climate variations; it may vary from one growing season to another, also depending on the grape variety, as noted by Chardonnet [37], Cabanne and Doneche [38], and Kidman et al. [39].

Ca accumulation in berry flesh and skin is affected by water supply and humidity (Figure 1A–C) and it is involved in plant stress response via cell wall growth and tissue development [5,38]. In general, the Ca content in the berry flesh increases steadily after the onset of flowering, and it reaches the highest value during the veraison, approximately 60 DAF. At the end of the veraison period, Ca accumulation in the pericarp decreases due to low mobility in the phloem and xylem rupture [9,39].

In spring 2019, more than in other growing seasons, high humidity delayed the onset of veraison while stimulating berry development and Ca accumulation in the berry flesh. In 2021, fairly similar curve patterns were observed for Ca accumulation in both flesh and skin, with the observation that the curves for the berry flesh were less representative than for the skin, especially in the ‘Merlot’ variety, probably due to the large data variability, which limits the definition of the regression curve. Numerous observations have indicated that Ca content does not normally increase after veraison [40,41,42,43]. There are also opinions that the Ca content increases throughout berry development, with accumulation mainly in seeds during the ripening stage [17,44,45,46]. Other researchers have opined that the Ca content is involved in cell expansion and turgor pressure during ripening [14]. In the cultivar ‘Chaunac’, Hrazdina et al. [12] reported a decrease in Ca accumulation during ripening. However, Possner and Kliewer [13] mentioned consistent Ca accumulation in ‘Chardonnay’ during ripening, especially in the skin. The migration of Ca from the flesh to the skin plays an important role in the process of tissue softening and the ripening of grapes [45,46]. Ca is involved in the water flux by modifying cell wall permeability, and thus it plays an important role in berry development [37] as well as resistance to pathogens [47]. Furthermore, Ca accumulation in berries depends on its mobility through the xylem, which is correlated with environmental conditions [16,48]. The decrease in Ca accumulation during the ripening stage in 2021 was influenced by the low temperature at night and the extremely hot and dry weather during daytime, which affected plant physiological responses. The decrease in xylem flow after the veraison stage corresponds to the shrinkage of berries due to the loss of cell vitality and the decrease in Ca import [41].

Differences between varieties regarding K accumulation after veraison were significant. K accumulation increased steadily both before and after veraison; similar behavior was observed by Possner and Kliewer [13] in ‘Chardonnay’ as well as by Rogiers et al. [17] in variety ‘Shiraz’. Different K levels in the berry skin of ‘Grenache Noir’ were also found to depend on moisture variations in different years, as mentioned by Etchebarne et al. [49], who also found that K accumulation is influenced by berry development and water supply. As shown in several studies [9,25,50], water deficiency leads to lower K concentrations in berries, as a result of this element’s mobility in soil and lower absorption by the roots [49,51], resulting in oxidative stress, which controls phloem transport and tissue turgor [52,53].

During ripening, the K accumulation rate was affected by water availability to the plants. Most of the Ca and K accumulation in the berries was observed before veraison. K continued to accumulate during and after veraison, while Ca accumulation was very slow. In high humidity conditions during the growing season of 2019, K accumulation was higher than in other seasons.

In the 2021 growing season, the weather in the study area was hot and dry (June—end of flowering and throughout the maturity phase in September–October). According to Villette et al. [52], when plants are under stress in drought conditions, their growth is impaired, and their K uptake capacity is reduced. Rogiers et al. [54] found a slow K accumulation of K before maturity in the variety ‘Shiraz’, followed by an increase in K content up to the point of maximum berry development. The same pattern was observed by Ollat and Gaudillère [45] in ‘Cabernet Sauvignon’ and Creasy et al. [14] in ‘Pinot Noir’.

The results of the studies conducted by Patrick and Offler [55] and Fontes et al. [56] indicated that the accumulation of K in the berries decreased towards the end of the ripening stage due to the decrease in phloem flow. Unlike Ca, K is mobile in both the xylem and phloem (being 10 times more mobile than in the xylem) [57,58]. The K content in berries is an important quality controller, especially in red wines and to a lesser degree in white wines, given that K is also extracted during the extraction of anthocyanins [59,60].

Cabanne and Doneche [7] and Rogiers et al. [54] reported that Ca accumulation in seeds increased during ripening in several grape varieties. Etchebarne et al. [49] found that K concentration in seeds of the grape variety ‘Grenache Noir’ continued to increase after veraison, while Ca accumulated in seeds mainly before veraison. Rogiers et al. [61] found that xylem flow before veraison supplied not only the flesh and skin of the berry but also the seeds; however, xylem flow after ripening was found to be restricted post-veraison to the central bundles and brush zone [14,61]. Rogiers et al. [54] reported similar results in variety ‘Shiraz’ (11 years old), where sugar content was correlated with K in a sigmoid pattern (R^2^ = 0.99, *p* < 0.01).

## 4. Materials and Methods

The studies were conducted in three consecutive growing seasons (2019–2021). Recaş Designation of Origin (DOC) is located in the Banat region of Western Romania, at the same longitude as Bordeaux and at an altitude between 170 and 220 m, GPS Coordinates: 45.826998, 21.5273451. The vines for aromatic and smooth wines are grown on an amphitheatre-like layout. The soil is clayey with limestone and sand and is iron-rich. The vineyard is characterized by a temperate continental climate with Mediterranean influences.

The ‘Cabernet Sauvignon’ and ‘Merlot’ vines from the Recaş vineyard, Timis County, were 11 and 9 years old, respectively. They were planted at a distance of 1.5 m between the vines and 2.5 m between the rows, oriented in a north–south direction and trained on simple Guyot, with 30 buds crop load. Berlandieri × Riparia SO4 rootstocks [62] were chosen for both varieties to ensure uniform nutrient uptake and more vigorous vines. The rootstock is resistant to phylloxera (*Daktulosphaira vitifoliae*) with good tolerance to nematodes (*Meloidogyne incognita, Meloidogyne arenaria*) and downy mildew (*Plasmopara viticola*) but is less tolerant to drought. The experimental field was designed with three replicates in a Latin square (4 × 4 vines/treatment) for each variety. ‘Cabernet Sauvignon’ and ‘Merlot’ varieties were selected for the study because they have similar berry sizes (13–14 mm diameter) and thick skin. The veraison period for both varieties is usually late July–early August, while ripening lasts from September to early October.

During the 2019 to 2021 seasons, 200 berries were harvested from each plot every 5 or 10 DAF until maturity then transported to the laboratory in plastic bags and frozen at −20 °C until analysis. Berries were collected at random from both sides of the row, from the top, bottom, front, and back of each cluster, while trying to ensure that they were roughly the same size. Because the Ca, K, and sugar accumulation in the berries were expressed in mg per berry, 50 berries were chosen and weighed (1.5 g ± 0.01 g), taking into account that the berries of both varieties usually weigh between 1 and 2 g. These 50 representative berries were separated into flesh and skin (seeds were removed), from which the juice was pressed and extracted separately, homogenized, and divided for component analysis (flesh, skin). Samples were homogenized using a one-hand control homogenizer (hand-held type, HMG-6) with a 10 G working head. For better homogenization, 0.5 mL of deionized water (DI water) was added to the seeds and skin samples. The samples were previously verified based on the size and density.

The Ca and K content in the juice extracted from the berries (flesh and skin) and seeds were determined using the flame atomic absorption spectroscopy (AAS) method, thus plotting a calibration curve. For Ca analysis, nine volumetric flasks of 100 mL each, numbered 1.0, 2.0, 3.0, 4.0, 5.0, 6.0, and 7.0 mL, were filled with 100 ppm Ca standard solution. A total of 1.0 mL of Sample 1 solution was added to the eighth flask, and 10.0 mL of Sample 2 solution was added to the ninth flask. All solutions were diluted to the appropriate concentration with distilled water. Ca concentration was calculated by comparing the standard curve after analyzing nine levels of Ca concentration (1 to 200 mg/L) in a solution diluted with distilled water at a wavelength of 422.7 nm; absorbance was measured three times for each solution. The Ca content was expressed in mg/berry after electrophoregrams were recorded and processed using the software Waters Millennium 2010 Chromatography Manager (Version 2.21; Waters Corporation, 34 Maple Street, Milford, MA 01757, USA). Three standard K solutions were prepared: 15, 30, and 45 ppm. The sample solutions were developed by diluting 1.0 mL of sample ‘1’ and 5.0 mL of sample ‘2’ solutions up to 100 mL. A K filter was installed on the instrument. The instrument was set to 0 with distilled water for relative emission intensity and to 100 with a 45 ppm solution. The relative emission intensity of the standard and sample solutions was measured. The K content was expressed in mg/berry after electrophoregrams were recorded and processed using the same software (Waters Millennium 2010 Chromatography Manager, Version 2.21).

Grape berries were crushed and the juice centrifuged at 4500× *g* for 5 min to determine the sugar content. The resulting supernatant was diluted four times with distilled water and stored at 40 °C for later sugar analysis. An automated microplate reader was used to measure sugar content enzymatically (ELx800UV, BioTek Instruments Inc., VT, USA). Climate data for the 2019–2021 growing seasons were taken from the weather station installed in the vineyard near the experimental blocks. Average minimum (av. min) temperatures were registered during the night (Table 1).

The total precipitation in the 2019 growing season (April–October) was 778.1 mm, which is significantly above normal (especially from April to July) and compared with the other two growing seasons. In contrast, the 2021 growing season was dry in June and September and extremely hot in July and August. Lower humidity was reported in the spring of the 2020 growing season, followed by heavy precipitation in the early and late veraison.

*Statistical analysis.* During the growing seasons (2019–2021), data were collected from ‘Cabernet Sauvignon’ and ‘Merlot’ varieties to analyze the skin and pulp of the berries for the evolution of Ca and K content (expressed in mg/berry), presented as second-degree polynomial regression (polynomial regression better describes the extreme values of the data; the curve shows the progress of the transition of berry development from one stage to another after flowering). XLSTAT Software 201 (vers. Microsoft Excel 2019, 16.0.6742.2048; Copyright Addinsoft Inc. 244 Fifth Avenue, Suite E100 New York, N.Y. 10001) was used to process the data (the significance of the terms was evaluated at alpha = 5%). The trend of Ca and K content, correlated with the number of DAF in ‘Cabernet Sauvignon’ and ‘Merlot’ varieties, was determined using a second-degree polynomial model. For the difference between the means (Ca and K content), the *t*-test for two independent samples (two-tailed test) was used. A multivariate statistical data analysis (MVA) of the samples was performed with one-way ANOVA (Prism 9 for Windows GraphPad Software, LLC. Version 9.3.1. (471); GraphPad Software2365 Northside Dr.Suite 560 San Diego, CA 92108). Electrophoregrams were recorded and processed using the software Waters Millennium 2010 Chromatography Manager (Version 2.21;Waters Corporation, 34 Maple Street, Milford, MA 01757, USA).

## 5. Conclusions

The effects of climate variability on grape berry development and components during the growing season are becoming increasingly evident, and some grape varieties are more sensitive to these changes than others. The results suggest that Ca and K accumulation in the skin, flesh, and seeds of grape berries are primarily determined by the stage of berry development. The results also show that Ca accumulation in the berry continues slowly after veraison; accumulation was observed mainly in the skin, indicating a shift from the flesh. The pattern of Ca and K accumulation in grape berries showed similarities in both components during ripening, and the highest accumulation was generally observed between 65 and 75 DAF in both varieties. The increase in K content as well as the increasing berry size may be related to water content in the berries during ripening. Understanding when the various components accumulate in grape berries may enable vineyard managers to plan fertilizer applications optimally, considering that deficiency even in a single nutrient may potentially decrease grape yield or quality. Therefore, the availability of each nutrient must be related to plant needs and administered by the winemaker at the right stages.

## Figures and Tables

**Figure 1 plants-11-01536-f001:**
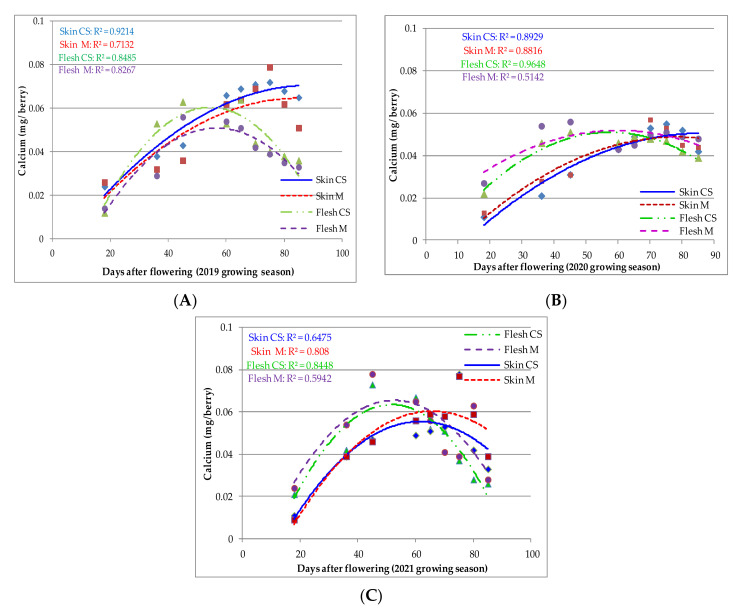
Ca accumulation in ‘Cabernet Sauvignon’ (CS) and ‘Merlot’ (M) berry skin and flesh during 2019–2021 growing seasons (**A**–**C**) from flowering to maturity. A scatterplot with a trend line of different colour and dash type was drawn to represent the relationships between DAF and Ca accumulation in flesh and skin for the CS and M varieties during 2019–2021 growing seasons. The regression analysis suggests that Ca accumulation changes with DAF. (
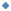
—Ca skin CS; 
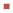
—Ca skin M; 
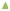
—Ca flesh CS; 
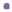
—Ca flesh M).

**Figure 2 plants-11-01536-f002:**
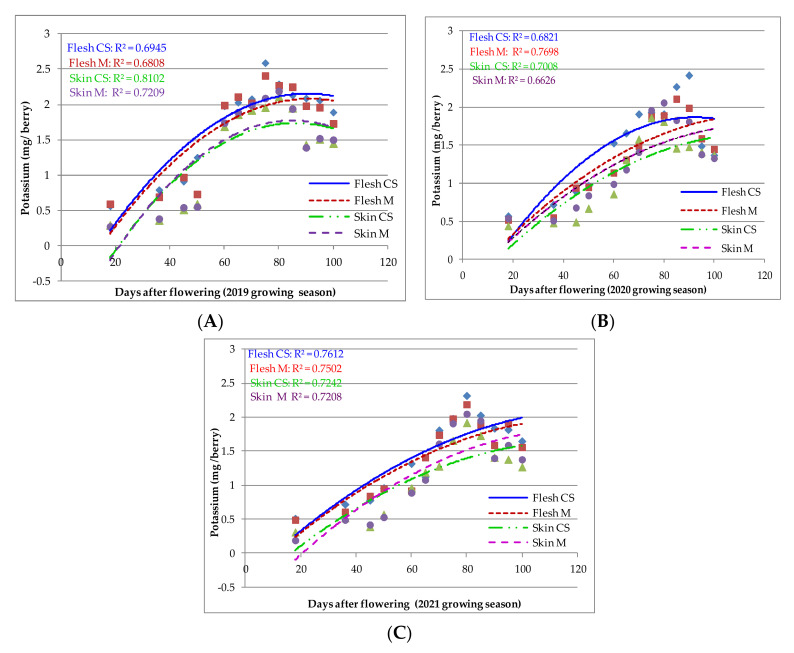
K accumulation in ‘Cabernet Sauvignon’ (CS) and ‘Merlot’ (M) berry skin and flesh during 2019–2021 growing seasons (**A**–**C**) from flowering to maturity. A scatterplot with a trend line of different colour and dash type was drawn to represent the relationships between DAF and K accumulation in flesh and skin for the CS and M varieties, during 2019–2021 growing seasons. The regression analysis suggests that K accumulation changes with DAF. (
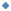
—K flesh CS; 
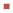
—K flesh M; 
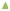
—K skin CS; 
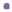
—K skin M).

**Figure 3 plants-11-01536-f003:**
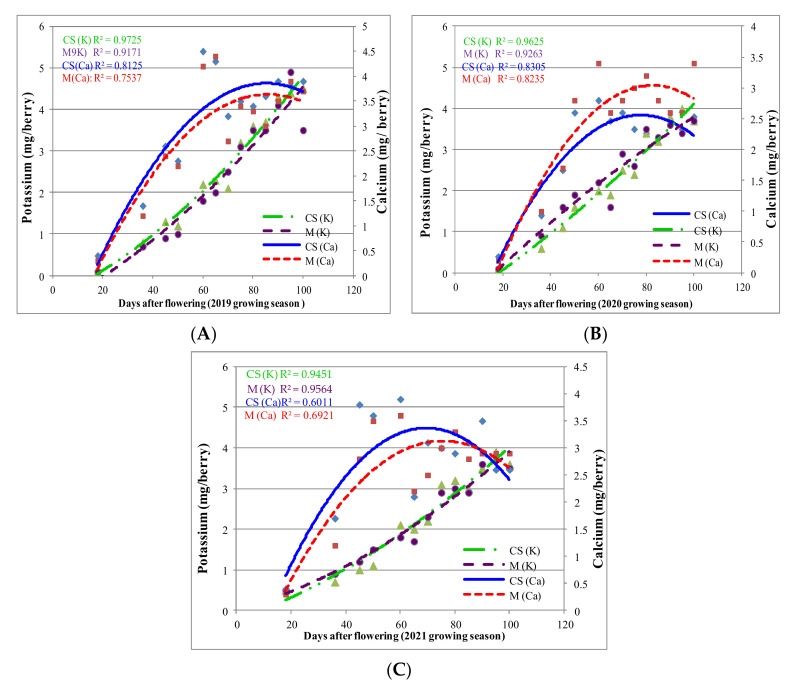
K and Ca accumulation in ‘Cabernet Sauvignon’ (CS) and ‘Merlot’ (M) berries during 2019–2021 growing seasons (**A**–**C**), from flowering to maturity. A scatterplot with a trend line of different colour and dash type was drawn to represent the relationships between DAF, Ca and K accumulation in the CS and M berries, during 2019–2021 growing seasons. The regression analysis suggests that Ca and K accumulation changes with DAF (
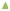
—K in CS berry; 
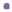
—K in M berry; 
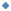
—Ca in CS berry; 
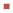
—Ca in M berry.

**Figure 4 plants-11-01536-f004:**
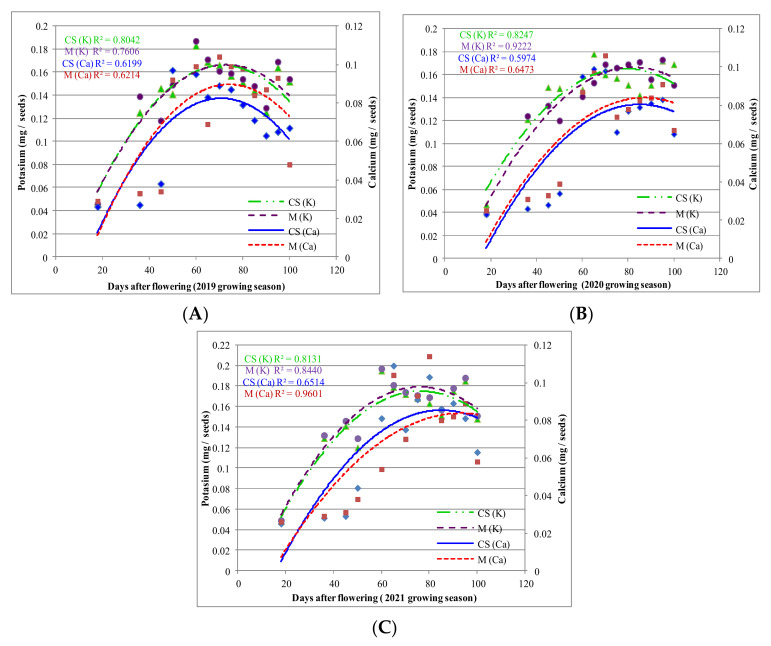
K and Ca accumulation in ‘Cabernet Sauvignon’ (CS) and ‘Merlot’ (M) berry seeds during 2019–2021 growing seasons (**A**–**C**), from flowering to maturity. To represent the relationships between DAF, Ca, and K accumulation in the CS and M seeds during the 2019–2021 growing seasons, a scatterplot with a trend line of different colour and dash type was drawn. The regression analysis suggests that Ca and K accumulation changes with DAF. (
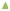
—K in CS seeds; 
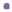
—K in M seeds; 
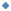
—Ca in CS seeds; 
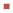
—Ca in M seeds).

**Figure 5 plants-11-01536-f005:**
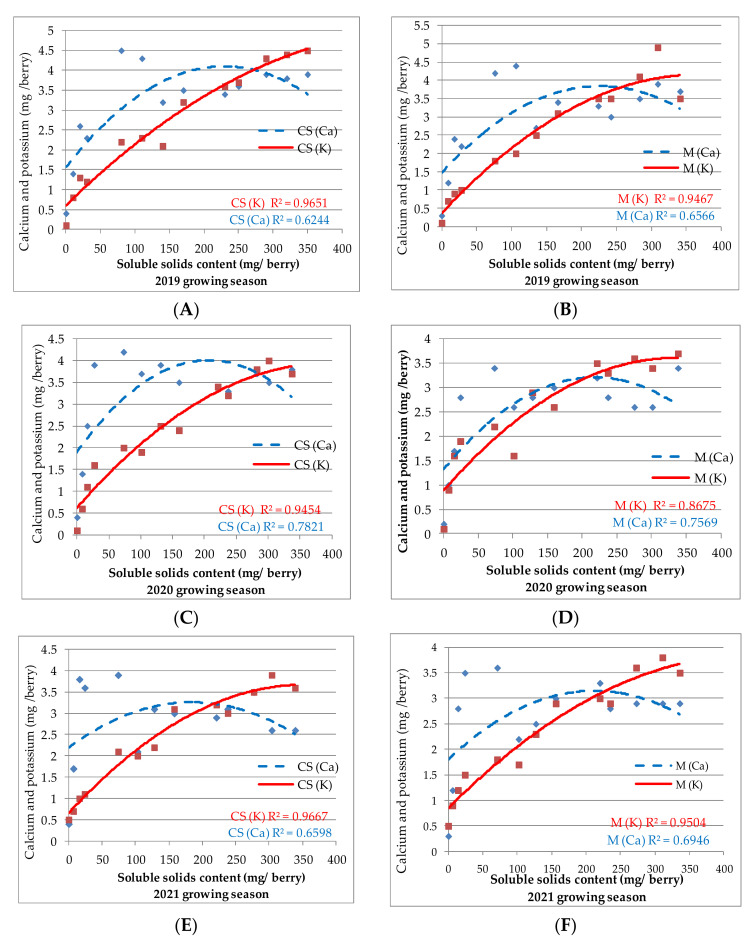
Sugar (soluble solids) accumulation and relationship with K and Ca in ‘Cabernet Sauvignon’ (CS) and ‘Merlot’ (M) berries in the 2019–2021 growing seasons, from flowering to maturity. To represent the relationships between DAF, sugar, Ca, and K accumulation in the CS and M berries during the 2019–2021 growing seasons, a scatterplot with a trend line of different colour and dash type was drawn. (**A**,**C**,**E**

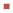
—K in CS berry; 
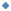
—Ca in CS berry; **B**,**D**,**F**

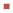
—K in M berry; 
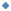
—Ca in M berry;).

**Table 1 plants-11-01536-t001:** Average monthly temperature and rainfall over three growing seasons (2019–2021): flowering (May-June), veraison (July-August), and harvest (September–October).

	2019			2020			2021		
	Av. Max (°C)	Av. Min (°C)	Total Rainfall (mm)	Av. Max (°C)	Av. Min (°C)	Total Rainfall (mm)	Av. Max (°C)	Av. Min (°C)	Total Rainfall (mm)
April	18.30	7.63	112.10	16.33	6.40	22.30	13.17	5.33	80.70
May	19.30	9.67	259.30	19.29	10.19	61.80	19.13	10.39	97.80
June	27.52	16.28	154.70	23.60	15.33	196.30	26.90	15.57	16.70
July	27.32	15.38	113.80	26.16	16.68	149.20	32.77	19.29	95.20
August	31.13	17.90	56.20	29.58	18.64	56.50	31.00	17.94	59.00
September	25.33	13.20	54.20	26.10	15.30	43.00	26.07	13.30	3.20
October	20.94	15.82	27.80	17.42	10.74	117.70	16.52	7.65	10.07

## Data Availability

Not applicable.

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
