# Peer review of "Calcium and Potassium Accumulation during the Growing Season in Cabernet Sauvignon and Merlot Grape Varieties"

_plants, 2022, doi:10.3390/plants11121536_

Round 1
Reviewer 1 Report
Review on manuscript ID: plants-1703389 “Calcium and potassium accumulation during growing season in Cabernet Sauvignon and Merlot grape varieties” by Eleonora Nistor, Alina Georgeta Dobrei, Giovan Battista Mattii and Alin Dobrei submitted to Plants
The article presented is well structured. The aim of studies is to clearly formulate and correctly select the analytical methods necessary for its implementation.
In my opinion the topic taken by Authors is interesting. Generally manuscript is readable , prepared correctly and could be publish in Plants after minor corrections.
Detailed recommendations:
Page 3, line 117
- There should be a full stop in the title at the end of the sentence.
Page 4, line 137
- There should be a full stop in the title at the end of the sentence.
Page 5, line 159
- There should be a full stop in the title at the end of the sentence.
Page 6, line 183
- There should be a full stop in the title at the end of the sentence.
Page 8 line 291
- There should be a full stop in the title at the end of the sentence.
Pages 9 - 11
In the reference list, the journal's volume number should be written in italics
Page 9, line 341
The page range should be given.

Author Response
Response:
Dear Ms/Mr,
First of all, thank you for your help in improving this paper.
In accordance with the requirements of the five reviewers, we have attempted to revise the manuscript, improving the translation and the content of each paragraph, including your requirements for comments on the pages and lines mentioned.
Thank you for your time!
Reviewer 2 Report
comments on attached file

Author Response
Response 1:
Dear Mr./Ms.,
First and above all, thank you for your assistance in improving this paper.
We attempted to revise the manuscript in accordance with the requirements of the five reviewers, improving the translation and content of each paragraph, as well as your requirements for comments on the pages mentioned.
Response 2:
A t-test analysis was performed to compare the mean values of the variables.
A second-degree polynomial function (general form: f(x)=Ax2+Bx+C) was used to identify the DAF with the highest calcium and potassium content in berry flesh, skin, and seeds.
Response 3:
For mg/berry seeds, we removed the word "berry" because it causes confusion.
DAF influenced the accumulation of calcium and potassium in skin and flesh in both varieties by more than 50% (R2 = 0.50). We specified the R2 value for each component in the diagrams because there is too much data to comment on.
Thank you very much for your time!

Reviewer 3 Report
General comments
The authors must explain well what their new contribution is. It is not clear what new results are compared to the state of the science.
Specific comments:
Point 2 must be Materials and methods instead of point 4.
The conclusions should be rewritten indicating the most relevant results of your investigation.
Lines 269-273: The text is in blue. Why?
Lines 301-303: Idem
Author Response
Dear Mr./Ms.,
First and above all, thank you for your assistance in improving this paper.
We attempted to revise the manuscript in accordance with the requirements of the five reviewers, improving the translation and content of each paragraph, as well as your requirements for comments on the pages mentioned.
Response 1:
The goal was to determine the days when the greatest accumulation of calcium and potassium occurs, depending on climatic parameters, in two varieties widely cultivated not only in Romania but also around the world, so that viticulturists could intervene with fertilizer application to produce the best wine possible.
Response 2:
Indeed, this is the format of most journals, but the format of the Plants Journal requires:
- Research manuscripts should comprise:
- Front matter: Title, Author list, Affiliations, Abstract, Keywords
- Research manuscript sections: Introduction, Results, Discussion, Materials and Methods, Conclusions (optional).
- Back matter: Supplementary Materials, Acknowledgments, Author Contributions, Conflicts of Interest, References.
Response 3:
We rewrote the conclusions.
Response 4:
We were also amazed that the text of the manuscript was colored! We have no explanation for these text color changes. But now we have colored in red all the changes we have made.
Thank you very much for your time!

Reviewer 4 Report
Although this work is a little beat out from my main skills, are interesting the obtained results and the form that the discussion is made. In overall the aim of work is interesting and the chapters are well presented and discussed but minor changes are required.
ln 91_ In the first decades of June with warmer weather, the vine began to recover from the delay in vegetation ... improve this phrase.
In all figures- example Ln 116 , use the same sequence for varieties, the same according colors. Example in front of Skin M: R² = 0.7760 put the red color Skin M . May be not this sequence but put this correspondence because improve the interpretation of data.
Ln 122 The correct word is flash or flesh?
Improve this phrase.
Ln 269-Berlandieri X Riparia SO4 rootstocks for uniformity include also one reference (VIVC https://www.vivc.de/index.php?r=passport%2Fview&id=11473
In All document put variety names in commas like ‘chardonnay’ please check according rules of plants journal
Ln 299 improves this phrase.
Ln 308 to 340 improve this phrase, is not clear.
Ln 322 check this phrase, where are these results in your work? Improve or describe on two phrases!
The references were not verified.
Good paper!
Author Response
Dear Mr./Ms.,
First and above all, thank you for your assistance in improving this paper.
We attempted to revise the manuscript in accordance with the requirements of the five reviewers, improving the translation and content of each paragraph, as well as your requirements for comments on the pages mentioned.
We also put in commas the name of both varieties.
We add one reference for rootstock.
We have no explanation for the manuscipt text color changes.
But now we have colored in red all the changes we have made.
Thank you for your time!

Reviewer 5 Report
The authors must take into account the following considerations to improve the submitted manuscript.
- Reorganize the content of the abstract following the structure proposed in the guide for authors.
- Review the bibliographic references used and incorporate updated information.
-In the results section, they must review the results of the influence of the climate with the data provided in table 1. Discrepancies have been observed.
-In the figures the units of measurement are mg/berry, this is imprecise. Do the berries of the two varieties have the same weight? Improve these graphics and not use colors to facilitate printing in black and white.
- Review the contents included in the results section. They include comments that are considered discussion. The discussion section should be revised. The authors should include in this section the discussion of their results.
- In the material and methods section, they must include the geographical coordinates of the plots and describe in detail how the sampling was carried out and the equipment used for the measurements.
- The authors should be very cautious when presenting the results since the varieties studied have different ages.
- Review the conclusions section. The authors present very general conclusions and this research work focuses only on two varieties.
Author Response
Dear Mr./Ms.,
First and above all, thank you for your assistance in improving this paper.
We attempted to revise the manuscript in accordance with the requirements of the five reviewers, improving the translation and content of each paragraph, as well as your requirements for comments on the pages mentioned.
Response 1:
We made the changes you suggested.
We updated the graphics. Because one of the reviewers requested that the trendline and the R2 values be kept in the same colors, we changed the style of the line so that it could be distinguished when printed in black and white.
Thank you for your time!

Round 2
Author Response
Thank you for assisting us in improving our manuscript and correcting errors.

Reviewer 3 Report
The requested changes have been made
Author Response
Thank you for your comments and assistance in improving the manuscript.
Reviewer 5 Report
The authors have not addressed all the changes proposed for the first version:
- Reorganize the content of the abstract following the structure proposed in the guide for authors.
- Review the bibliographic references used and incorporate updated information
- In the figures the units of measurement are mg/berry, this is imprecise. Do the berries of the two varieties have the same weight?.
- In the material and methods section: Authors should describe the equipment used.
- The authors should be very cautious when presenting the results since the varieties studied have different ages.
Lines 93-94: “In 2019, the spring was cold, especially overnight, delaying the vines' vegetative and reproductive development by about two weeks”. This information is not reflected in Table 1. The results must be evidenced with data. Average night temperatures are not included.
The acronyms must indicate what it means the first time, for example DAF
Author Response

(The authors gave the same response as above.)

Round 3
Author Response
"Please see the attachment."

Reviewer 5 Report
There is information in the manuscript that should be clarified.
1. In the figures the units of measurement are mg/berry. Authors should indicate the weight of the berry.
2. In the material and methods section, they must indicate in detail the equipment used for the different analyses. It is essential to describe the methodology and equipment used in order to reproduce the study in other laboratories that may be interested.
Author Response
"Please see the attachment".
